# The Antiviral Effects of Jasminin via Endogenous TNF-α and the Underlying TNF-α-Inducing Action

**DOI:** 10.3390/molecules27051598

**Published:** 2022-02-28

**Authors:** Xiaohong Zhu, Ziwei Hu, Tian Yu, Hao Hu, Yunshi Zhao, Chenyang Li, Qinchang Zhu, Mingzhong Wang, Peng Zhai, Longxia He, Muhammad Shahid Riaz Rajoka, Xun Song, Zhendan He

**Affiliations:** 1Affiliated Longhua People’s Hospital, Southern Medical University, Shenzhen 518172, China; szlhyyjxk2021@126.com; 2School of Pharmaceutical Sciences, School of Basic Medicine, Health Science Center, Shenzhen University, Shenzhen 518000, China; huziwei2019@163.com (Z.H.); bobodeyutian@163.com (T.Y.); huhaohh11@163.com (H.H.); zhaoyunshi@szu.edu.cn (Y.Z.); lcy@szu.edu.cn (C.L.); 3College of Pharmacy, Shenzhen Technology University, Shenzhen 518118, China; zhuqinchang@sztu.edu.cn (Q.Z.); wangmingzhong@sztu.edu.cn (M.W.); 4Faculty of Information Technology, Macau University of Science and Technology, Macau 999078, China; 1909853gii30004@student.must.edu.mo; 5Department of Otorhinolaryngology-Head and Neck Surgery, Chengdu Integrated TCM&Western Medicine Hospital, Chengdu 610017, China; npcr417@126.com; 6Laboratory of Animal Food Function, Graduate School of Agricultural Science, Tohoku University, Sendai 980-8572, Japan; shahidrajoka@yahoo.com

**Keywords:** jasminin, TNF-α, RAW264.7 cells, MAPKs, antiviral, HSV-1

## Abstract

Previous studies have reported that recombinant tumor necrosis factor (TNF)-α has powerful antiviral activity but severe systematic side effects. Jasminin is a common bioactive component found in Chinese herbal medicine beverage “Jasmine Tea”. Here, we report that jasminin-induced endogenous TNF-α showed antiviral activity in vitro. The underlying TNF-α-inducing action of jasminin was also investigated in RAW264.7 cells. The level of endogenous TNF-α stimulated by jasminin was first analyzed by an enzyme-linked immunosorbent assay (ELISA) from the cell culture supernatant of RAW264.7 cells. The supernatants were then collected to investigate the potential antiviral effect against herpes simplex virus 1 (HSV-1). The antiviral effects of jasminin alone or its supernatants were evaluated by a plaque reduction assay. The potential activation of the PI3K–Akt pathway, three main mitogen-activated protein kinases (MAPKs), and nuclear factor (NF)–κB signaling pathways that induce TNF-α production were also investigated. Jasminin induces TNF-α protein expression in RAW264.7 cells without additional stimuli 10-fold more than the control. No significant up-expression of type I, II, and III interferons; interleukins 2 and 10; nor TNF-β were observed by the jasminin stimuli. The supernatants, containing jasminin-induced-TNF-α, showed antiviral activity against HSV-1. The jasminin-stimulated cells caused the simultaneous activation of the Akt, MAPKs, and NF–κB signal pathways. Furthermore, the pretreatment of the cells with the Akt, MAPKs, and NF–κB inhibitors effectively suppressed jasminin-induced TNF-α production. Our research provides evidence that endogenous TNF-α can be used as a strategy to encounter viral infections. Additionally, the Akt, MAPKs, and NF–κB signaling pathways are involved in the TNF-α synthesis that induced by jasminin.

## 1. Introduction

Herpes simplex viruses, also known as HSVs, are double-stranded linear DNA viruses that cause viral infections in humans [1]. There are two genetically and serologically distinct human HSVs, HSV-1 and HSV-2 [1]. HSV-1 causes contagious orofacial infections and rarely encephalitis in newborns and infants [1]. An estimated 20−33% of children experience HSV-1 infection by age five via oral herpes [2]. HSV-1 can cause keratitis and encephalitis among immunocompromised HIV-infected individuals [2]. First-line antiviral drugs such as famciclovir, acyclovir, and valacyclovir are routinely used to treat HSV-1 infections. Although this medication can modulate the course of the disease, it cannot cure the infection. Additionally, there is no vaccine for HSV-1. Therefore, there is an urgent need to explore new and effective strategies against HSV-1 infections. Antiviral cytokines such as interferons (IFNs), interleukins (ILs), and tumor necrosis factors (TNFs) perform both the induction and regulation of innate and adaptive antiviral mechanisms when viruses infect the host. Based on the antiviral characteristics of cytokines, they are used as immune inducers or drugs to treat persistent viral infection.

Tumor necrosis factor-α (TNF-α) is an inflammatory cytokine produced by macrophages and is responsible for inflammation, immunity, and important for resistance to infection and cancers [3]. Moreover, recombinant TNF-α has potent, direct antiviral activity but has not been exploited as an antiviral therapy because of its systemic toxic effects [4,5]. Research has shown that exogenously recombinant TNF-α has a protective effect in mouse models of HSV-1 infection [6]. Lundberg et al. reported that recombinant TNF-α protects resistant C57BL/6 mice against HSV-1 induced encephalitis through TNF receptors [7]. The endogenous TNF-α also has an essential role in protecting viral infection; therefore, some systemic toxicity may be avoided [8]. 

Multitude of natural products are known to regulate endogenous cytokine production [9,10,11,12,13]. Jasminin is a secoiridoid glucoside isolated from the leaves of *Jasminum nudiflorum* Lindl., which is commonly used in the folk medicine of many Asian countries. In China, the leaves of *J. nudiflorum* are used as medicinal herbs for treating many disorders, including influenza, fever, and wound-healing [14]. However, very few studies have reported the biological effects of jasminin. Thus, our current research studies the antiviral effects of jasminin, one of the major constituents of *J. nudiflorum*, which may help explain its use in folk medicine.

## 2. Results

### 2.1. Jasminin Can Activate RAW 264.7 Cells and Induce TNF-α Production

Activated macrophages have a defensive function against viral infection; thus, macrophage activation by jasminin was investigated. As shown in Figure 1B, significant increases in cell proliferation were observed at all tested concentrations of jasminin from 3.1 to 200 μΜ, and even 3.1 μM of jasminin resulted in 420% cell proliferation compared with control.

Classically activated macrophages are accompanied by the release of various cytokines [15]. In order to understand whether jasmine affects the production of cytokines, cytokines in cell culture supernatants were quantified by ELISA. As shown in Figure 1C, jasminin (3.1–200 µM) induced TNF-**α** production in a dose-dependent manner in RAW 264.7 cells. The maximum TNF-α released in RAW 264.7 cells were about 500 pg/mL at a concentration of 50 μM, and basal TNF-α production was considered to be about 30 pg/mL. Lipopolysaccharides (LPS), as a positive control, induced TNF-α release to 600 pg/mL. In addition, none of type I IFNs were affected by jasminin stimulation in RAW 264.7 macrophages (Appendix A).

### 2.2. Endogenous TNF-α Induced by Jasminin Has Potent Antiviral Activity against HSV-1

The antiviral effect of jasminin was evaluated by a plaque reduction assay against HSV-1. At the concentration range of 6.25 to 200 μM, jasminin and cell culture supernatants collected from jasminin-treated RAW 264.7 cells are nontoxic to Vero cells (Figure 2a). As shown in Figure 2b, jasminin has no antiviral effects at the concentrations of 6.25 and 12.5 μM. Mild antiviral effects against HSV-1 are observable at 25 and 50 μM. Interestingly, jasminin treated cell culture supernatants containing endogenous TNF-α protect Vero cells against HSV-1, whereas stimulation by jasminin alone does not induce significant antiviral activity at the concentrations of 6.25 and 12.5 μM (Figure 2b,c). In detail, the plaque number of HSV-1 was significantly decreased (plaque numbers for cell culture supernatants from stimulation of jasminin at 6.25, 12.5, 25, and 50 μM: 40.33 ± 4.72, 32.00 ± 1.73, 30.00 ± 1.00, and 25.00 ± 2.00, respectively, versus 49.00 ± 2.00 in mock-treated cultures; *p* < 0.05) after being treated with cell culture supernatants in Vero cells (Figure 2d). The plaque size of HSV-1 also decreased (cell culture supernatants from stimulation of jasminin at 25 and 50 μM: diameters of 0.16 ± 0.08 and 0.13± 0.07 mm, respectively, versus 0.21 ± 0.08 mm in mock-treated cultures) (Figure 2d). These results suggest that endogenous TNF-α has strong anti-HSV-1 effects; furthermore, the endogenous TNF-α that was induced by jasminin displayed indirect antiviral effects in vitro.

### 2.3. Akt Phosphorylation Mediates TNF-α Production Induced by Jasminin

It has been reported that the phosphatidylinositol-3-kinase (PI3K)/Akt signaling pathway is involved in regulating TNF-α production [16]. Thus, the role of PI3K–Akt in jasminin-induced signaling pathways was further investigated through the phosphorylation of Akt, which was a key downstream effector of PI3K. As shown in Figure 3, Akt phosphorylation was increased at 5 min in RAW 264.7 cells after treatment with 50 μM of jasminin. Additionally, increased phospho-Akt levels persisted until 30 min after treatment and gradually declined afterward. 

Pretreatment of RAW 264.7 cells with LY294002, a specific PI3K–Akt inhibitor, 1 h before jasminin challenge (50 μM) was effective to block jasminin-provoked TNF-α production in a dose-response manner (Figure 3C). These data implicate an important role for PI3K/Akt signaling in jasminin-induced TNF-α expression.

### 2.4. Phosphorylation of MAPKs by Jasminin to Induce TNF-α Production

MAPKs-related signaling pathways are involved in TNF-α production in RAW264.7 cells. In Figure 4, ERK1/2, JNK1/2, and p38 MAPK activation by jasminin in RAW 264.7 cells were observed via Western blotting. We found that jasminin at 50 μM stimulated rapid and strong phosphorylation of all three MAPKs in a time-dependent manner. In parallel, we observed that jasminin induced both JNK and p38 kinase activation at 5 min and ERK phosphorylation at 15 min, and all three subfamilies of MAPKs reached their maximal phosphorylation levels at 15 min and declined thereafter (Figure 4A). 

To verify whether the activation of MAPKs by jasminin enhances TNF-α release, RAW264.7 cells were pretreated with specific JNK, ERK and p38 inhibitors, including SP600125, PD98059, and SB203580 respectively, for 30 min before incubation with 50 μM jasminin. The observed results in Figure 4C revealed that the inhibitors significantly attenuated the induction of TNF-α by jasminin in a dose-response manner. About 90% of TNF-α secretion was blocked when SP600125 (30 μM) was applied. In parallel, the ERK and p38 MAPK inhibitors PD98059 and SB203580 were less potent in inhibiting jasminin-stimulated TNF-α secretion than SP600125, as displayed in Figure 4C. It was of interest to ascertain that the observed delay in jasminin-induced MAPK activation was paralleled by a delay in TNF-α secretion. Figure 4C shows that this was the case because significant TNF-α elevations were first detected at 1 h after the application of jasminin (Appendix A).

### 2.5. NF–κB Signal Transduction Induced by Jasminin in RAW 264.7 Cells

The NF–κB family of transcription factors has an essential role in TNF-α secretion. Jasminin activates the MAPKs, which activate the classical pathway of NF–κB transcription factors including NF–κB (p65) and I-κBα, which coordinate the expression of its responsive cytokine TNF-α. As shown in Figure 5A, when the cells were additionally treated with jasminin, the nuclear translocation was activated. However, pretreatment with the NF–κB inhibitor for 1 h further significantly suppressed NF–κB p65 nuclear translocation via immunofluorescence analysis. We further observed that jasminin activated the NF–κB signal pathway in a time-dependent way by Western blot analysis (Figure 5B), which indicated that jasminin-induced TNF-α secretion is partly due to its NF–κB-activation effects.

As previously reported, Bay-11-7082 (BAY) is a strong inhibitor of NF–κB activity [17]. Thus, BAY was used to confirm whether jasminin stimulate TNF-α release via NF–κB activation in RAW 264.7 cells. Our results (Figure 5D) revealed that BAY almost blocked the secretion of TNF-α in the jasminin-treated RAW 264.7 macrophages at 3 μM, indicating that jasminin was involved in the increased NF–κB activity and, hence, TNF-α production. Furthermore, a jasminin-dependent increase of NF–κB-activity was confirmed by directly observing NF–κB levels in the cells. 

## 3. Discussion

TNF-α was initially identified as an antitumor cytokine against a wide variety of tumor types [18]. Recombinant protein technology is widely used to reveal the multiple roles of TNF-α, such as inflammatory responses, antiviral, and antimicrobial effects [19,20]. The antiviral cytokine of IFN-β can be induced by TNF-α in several cell lines [21,22]. Previous studies showed that TNF-α and IFNs exert synergistic antiviral actions when replicating hepatitis C virus, hepatitis E virus [23], respiratory viruses [24], and poxviruses [25]. TNF-α also plays a pivotal role in the synergistic antiviral effect that IFN-γ has against cytomegalovirus (CMV) infection [26]. TNF-α also inhibits the replication of the Sin Nombre virus [27], swine influenza virus, and porcine respiratory coronavirus [28]; the vesicular stomatitis virus (VSV) [29]; and HSV-1 and HSV-2 [30]. The application of recombinant TNF-α (50 ng/mL) may decrease the accumulation of virus nucleocapsid protein in vitro [27]. Lane et al. demonstrated that the exposure of HIV-1 infected peripheral blood monocytes and macrophages to TNF-α (50 ng/mL) significantly inhibited viral replication [31]. Adding 10,000 ng of TNF-α in human lung carcinoma cells significantly suppressed replications of VSV, encephalomyocarditis virus (EMCV), adenovirus, and HSV-2 from 10- to 100-fold [32]. Recombinant TNF-α has similar biological activity with endogenously TNF-α, but administration of recombinant TNF-α to patients is always accompanied by severe toxic side effects such as fever, headache, nausea, vomiting, and hypotension [33,34,35], possibly due to a short half-life (15–30 minutes), low bioavailability, and purities [36]. In contrast, endogenously produced TNF-α may serve as a potential therapeutic drug to fill the gaps. Meanwhile, the safety of endogenously TNF-α is dependent on the toxicity and inducibility of the stimuli. Beforehand, whether endogenous TNF-α induced by cells has antiviral effects must be addressed. 

This study showed that RAW 264.7 cells under in vitro conditions were activated by jasminin to produce TNF-α, whereas no induction of IFNs was observed. RAW 264.7 cell density affects the expression levels of TNF-α at the same concentration of jasminin. Elevated expressions of TNF-α in Figure 3, Figure 4 and Figure 5 were observed compared to that in Figure 1 due to the increased cell density. Interestingly, jasminin-induced endogenous TNF-α exhibited enhanced antiviral effects in HSV-1 infected Vero E6 cells. However, the mechanisms by which jasminin significantly increases TNF-α production have not been elucidated. It has been postulated that classical PI3K/Akt-, MAPKs-, and NF–κB-activated pathways and mechanisms were involved in endogenous TNF-α production.

PI3K/AKT signal pathways can mediate inflammatory reaction via activating NF–κB and Nrf-2 [37]. PI3K/Akt signaling is required to activate MAPK and NF–κB signaling. Jasminin may significantly promote Akt activity in RAW264.7 cells, suggesting that MAP kinase may be activated based on increased Akt activity. Activating MAP kinases such as ERK, JNK, and p38 MAP kinases can lead to TNF-α secretion [38,39]. The results of our study revealed novel information about jasminin-activated signaling pathways associated with TNF-α overproduction in RAW 264.7 cells. Jasminin caused the degradation of IκB-α protein and activated MAP kinase in RAW 264.7 cells, which were associated with an increase in TNF-α protein expression. Our results indicated that all three major MAP kinases were activated by jasminin in alterations of TNF-α production.

Cytoplasmic NF–κB is an inactive complex with an inhibitory subunit IκB-α in RAW 264.7 cells [40]. Phosphorylation of IκB-α activates NF–κB, thereby rendering NF–κB proteins free to translocate into the nucleus and interact with the promoter–enhancer region of the target TNF-α genes [41]. NF–κB activation was evaluated by detecting IκB-α degradation and p65 up-expression. As expected, jasminin caused a rapid degradation in cytoplasmic IκB-α protein levels associated with the up-expression of p65 in 30 min, indicating that jasminin activated NF–κB pathway in RAW 264.7 cells. 

Finally, we used specific inhibitors to verify the activations of MAPKs, NF–κB, and/or PI3K/Akt pathways for inducing TNF-α production in RAW 264.7 cells. We demonstrated that jasminin-induced TNF-α productions significantly decreased with inhibitors of MAPKs, NF–κB, and PI3K/Akt pathways in RAW 264.7 cells. The data demonstrated that MAPKs, NF–κB, and PI3K/Akt pathways are involved in TNF-α production in jasminin-stimulated macrophage cells (Figure 6). These phosphorylation events trigger the TNF-α production, which was detected with a delay of 30–60 min in jasminin-stimulated RAW264.7 cells. The possible reason for the delay could be the phosphorylation events are in the early signal transduction mechanisms. Our study is also the first to report that jasminin-stimulated TNF-α production was positively regulated by MAPKs, NF–κB, and PI3K/Akt in RAW 264.7 cells.

## 4. Materials and Methods

### 4.1. Virus Strains

Jinan University in Guangzhou, China, kindly donated laboratory-adapted HSV-1 stocks. The HSV-1 stock was propagated on Vero E6 cells. The cell suspension was then harvested and frozen for future use. A total of 50% tissue culture infective dose (TCID_50_) assay was used to determine virus titration [42].

### 4.2. Cell Culture and Reagents

RAW 264.7 and Vero E6 cells were purchased from ATCC (ATCC TIB-71 and ATCC CRL-1586, respectively). Jasminin (purity ≥ 98%) was purchased from Yunshan Bio-Tech (Guangzhou, Guangdong, China).

### 4.3. Cell Proliferation

A 3-(4,5-dimethylthiazol-2-yl)-2,5-diphenyltetrazolium bromide (MTT) assay was adopted to evaluate the effects of cell proliferation by jasminin and cell supernatants. A total of 100 μL of Vero E6 or RAW 264.7 cells were cultured in 96-well tissue culture plates at a seeding density of 5 × 10^5^ viable cells/mL in DMEM medium with 10% fetal bovine serum (FBS) for 24 h. Then, 100 μL of jasminin (final concentrations at 3.1, 12.5, 50, and 200 μM) or its supernatant were added for another culture of 24 h. Cells were then treated with MTT reagent (10 μL/well, 5 mg/mL) for 4 h at 37 °C. The medium was gently removed, and 100 μL of DMSO was used to dissolve formazan crystals. The absorbance values were recorded at 570 nm using a microplate reader to determine cell proliferation.

### 4.4. In Vitro Stimulation for Cytokines Production

RAW 264.7 cells (1 × 10^4^ cells/mL, 200 μL/well) were cultured with serial dilutions of jasminin for 24 h, followed by collection of supernatants for cytokines detection by ELISA (enzyme-linked immunosorbent assay). ELISAs were performed according to the manufacturer’s instructions. 

### 4.5. Plaque Reduction Assay

RAW 264.7 macrophages at 5000 cells/well were loaded onto plates and were divided into five groups, including the following: culture medium with DMSO (negative control group) and culture medium supplemented with different concentrations of jasminin (6.25, 12.5, 25, and 50 μM of jasminin groups). Culture supernatants were collected at 24 h after above treatments for plaque reduction assay. 

A plaque reduction assay was performed to assess the antiviral effects of jasminin and supernatants. Dilutions of HSV-1 at 1 × 10^3^ pfu/mL were inoculated on Vero E6 cells and cultured for 2 h. After adsorption, we removed the cell supernatant and overlaid the cells with DMEM containing 1.2% methylcellulose in the presence of jasminin alone or RAW 264.7 cell supernatant stimulated by jasminin at four different concentrations (6.25, 12.5, 25, and 50 μM) separately. Each sample was examined in triplicate. Cells were fixed at 3 days of postinfection and stained with crystal violet, and plaque detection was performed with an inverted microscope (Leica, Wetzlar, Hessen, Germany). 

### 4.6. Western Blotting Analysis

RAW 264.7 cells (1 × 10^5^ cells/well) were parallelly cultured in 24-well plates with 50 μM jasminin or inhibitors for 24 h, and the cell culture supernatants were then collected for TNF-α detection. The parallel cultured plates were treated with jasminin (50 μM) or inhibitors for 0–60 min, followed by cell lysis with RIPA buffer containing 1 mM PMSF (Beyotime Biotechnology, Shanghai, China). The extracted proteins were then subjected to SDS-PAGE/immunoblot analysis using antiphospho- and antitotal ERK1/2 antibodies, antiphospho- and antitotal p38 antibodies, antiphospho- and antitotal JNK1/2 antibodies, and anti-β-actin antibody (all from Cell Signaling Technology, Beverly, MA, USA). The integrated density values of the Western blot bands were measured with a Chemiscope Western Blot Imaging System (Clinx ChemiScope, Shanghai, China).

### 4.7. Immunofluorescence Assay

RAW 264.7 cells were cultured overnight at 37 °C; then, the cells were washed, recultured in fresh medium with 50 μM of jasminin, vehicle control, or combined 50 μM of jasminin and 3.0 μM BAY 11-7082. After 24 h of incubation, cells were fixed by 4% formaldehyde on ice, followed by permeabilization with 0.25% Triton X-100 (Bio-Rad, Hercules, CA, USA). Cells were then blocked with 5% BSA for 1 hour, followed by incubation with NF–κB P65 antibody (BD Bioscience, Palo Alto, CA, USA). After 12 h of incubation at 4 °C, the cells were washed and bathed with the antirabbit Cy3^®^ secondary antibody (Abcam, Boston, MA, USA) for 1 h at 25 °C. After the second wash, the cells were stained with 5 ng/mL DAPI (Sigma-Aldrich, St. Louis, MO, USA). Pictures were captured under an DMI6000B confocal microscope (Leica Microsystems, Wetzlar, Hessen, Germany). The translocation of NF–κB in the nucleus was performed by obtaining the mean fluorescence intensity.

### 4.8. Statistical Analysis

Each assay was independently performed at least three times, and data were presented as mean ± SD. The results were analyzed by SPSS 17.0 software using Duncan’s Multiple Range test following one-way ANOVA. 

## 5. Conclusions

Collectively, our study provides a novel role for endogenous TNF-α induced by jasminin in inhibiting HSV-1 replication. This antiviral effect of jasminin is mediated via endogenous TNF-α through the classical PI3K/Akt, MAPKs, and NF–κB signaling pathways. Furthermore, this finding may contribute to a novel understanding of the endogenous TNF-α usage in viral infection.

## Figures and Tables

**Figure 1 molecules-27-01598-f001:**
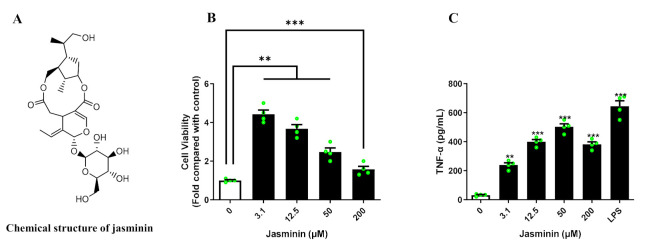
(**A**) The chemical structure of jasminin. (**B**) Jasminin enhances the proliferation of RAW 264.7 cells. (**C**) RAW264.7 cells treated with jasminin (3.1, 12.5, 50, and 200 μM) or LPS (1 μg/mL) for 24 h. The medium was collected to determine the TNF-α level using ELISA. Values are expressed as mean ± SD, *n* ≥ 3. ** *p* < 0.01 and *** *p* < 0.001 vs. control (jasminin-untreated).

**Figure 2 molecules-27-01598-f002:**
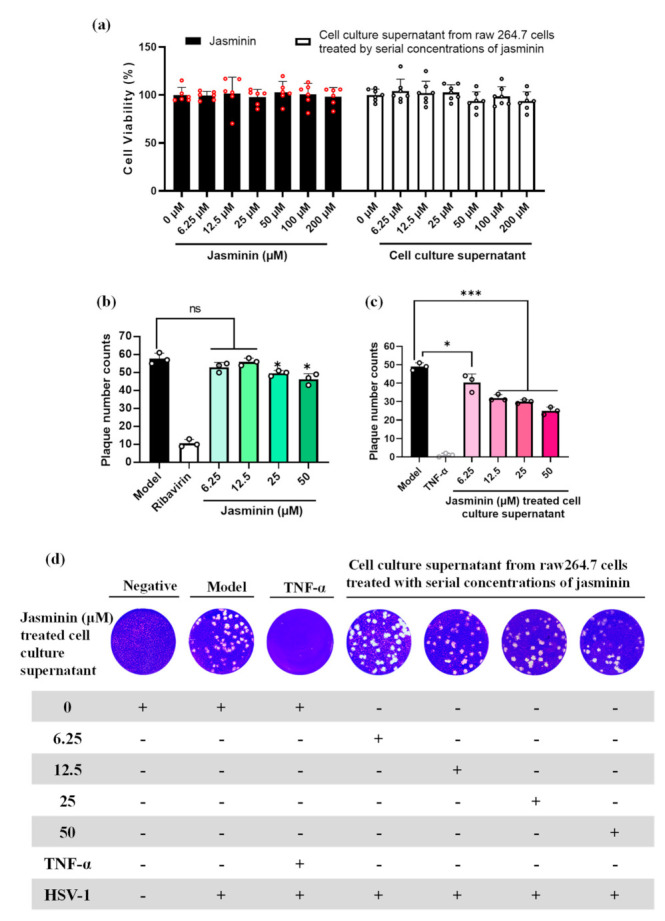
The plaque reduction assay of jasminin and RAW 264.7 cell culture supernatants stimulated by jasminin to inhibit HSV-1. (**a**) Cytotoxicity test on Vero E6 cells for jasminin alone or supernatants from RAW 264.7 cells treated by jasminin. (**b**) Antiviral effects of jasminin alone on Vero E6 cells infected with HSV-1. Ribavirin (20 μg/mL) served as a positive control. (**c**) Antiviral activity of RAW 264.7 cell culture supernatant stimulated by jasminin on HSV-1. RAW264.7 cells were treated with various concentrations of jasminin (6.25, 12.5, 25, and 50 μM) for 24 h. Recombinant TNF-α (100 pg/mL) was served as a positive control. (**d**) Crystal violet-stained Vero E6 cell monolayers showing plaques generated by HSV-1 at 72 h postinfection. For each well, jasminin cell culture supernatants were finally 10-fold diluted. Experiments were performed in triplicate, and results are shown as the mean ± SD. * *p* < 0.05 and *** *p* < 0.001 compared to control.

**Figure 3 molecules-27-01598-f003:**
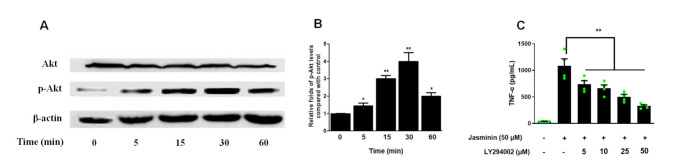
Jasminin induces Akt phosphorylation in mouse macrophages. (**A**) RAW 264.7 cells were treated with jasminin (50 μM) for 0–60 min. Whole-cell lysates were then prepared for Western blot analysis of phospho-Akt (Ser-473) expression at each time point. (**B**) Densitometry of phospho-Akt expression relative to Akt. (**C**) Effects of PI3K inhibitor on TNF-α protein production induced by jasminin. RAW264.7 cells were treated with inhibitors (0 to 30 μM) in the presence or absence of jasminin (50 μM), respectively. The levels of TNF-α (**C**) in the cell culture supernatants were measured by ELISA. Results are displayed as means ± SEM from at least three independent experiments. * *p* < 0.05/** *p* < 0.01 vs. control (B, time = 0 min) or cells treated with jasminin alone (**C**).

**Figure 4 molecules-27-01598-f004:**
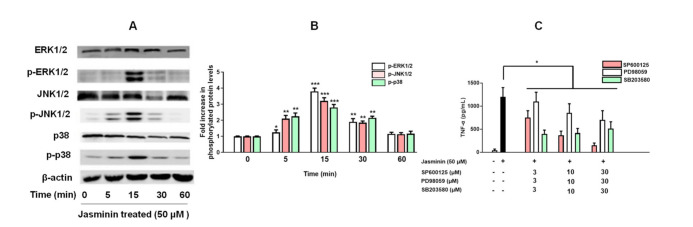
Time course of MAPK phosphorylation in RAW 264.7 cells stimulated with jasminin. (**A**) RAW 264.7 cells were treated with jasminin alone for 0, 5, 15, 30, and 60 min. (**B**) Western blot analysis was performed of phosphorylated p38, JNK1/2, and ERK1/2, respectively, and normalized vs. total p38, JNK1/2, and ERK1/2 protein, respectively. (**C**) Effects of MAPK inhibitors on TNF-α protein release induced by jasminin. Levels of TNF-α by ELISA from the cell culture supernatant of RAW264.7 cells treated with inhibitors (0−30 μM) in the presence or absence of jasminin (50 μM). * *p* < 0.05, ** *p* < 0.01 and *** *p* < 0.001 compared to control. The immunoblotting results represent three independent experiments.

**Figure 5 molecules-27-01598-f005:**
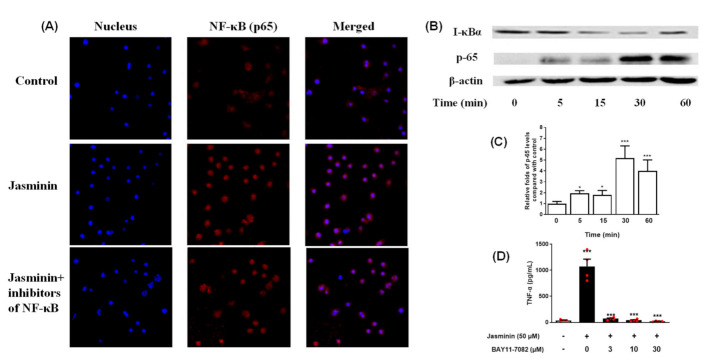
Jasminin-induced NF–κB activation in RAW 264.7 cells. (**A**) Cells were treated with jasminin for 1 h. The localization of NF–κB p65 was visualized with fluorescence microscopy after immunofluorescence staining with anti-NF–κB p65 antibody (red). The cells were also stained with DAPI to visualize the nuclei (blue). (**B**) Western blot test of p65 and I-κBα of RAW 264.7 treated by jasminin for 0–60 min. (**C**) Fold increase of phospho-65 expression is shown. (**D**) Effect of BAY 11-7082 on TNF-α protein production induced by jasminin. The data are presented as the mean ± SEM (*n* ≥ 3, * *p* < 0.05, *** *p* < 0.001 compared to control without both jasminin or BAY 11-7082, ANOVA).

**Figure 6 molecules-27-01598-f006:**
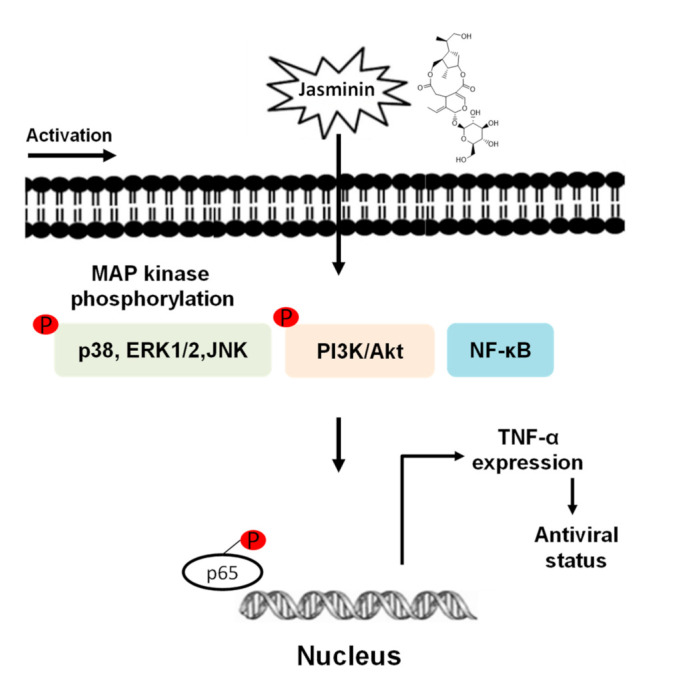
Jasminin-induced signaling pathways involved in TNF-α production in macrophages.

## Data Availability

All data generated or analyzed during this study.

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
