# Peer review of "The Antiviral Effects of Jasminin via Endogenous TNF-α and the Underlying TNF-α-Inducing Action"

_molecules, 2022, doi:10.3390/molecules27051598_

Round 1
Reviewer 1 Report
The antiviral effects of jasminin via endogenous TNF-α and 2 the underlying TNF-α-inducing action
The submitted research article investigates antiviral affects mediated by jasminin, a natural product used in traditional medicine application in Asian countries. The investigators have identified that TNF-alpha expression is augmented in jasminin stimulated macrophages and it imparts anti-viral activities.
The findings are relevant to the scope of the journal. However, the manuscript, at the current state cannot be recommended for publication as it needs to undergo major revisions.
- Overall, the language use in the manuscript is poor with many grammatical errors. I would suggest authors to obtain the assistance from an editorial service to revise the manuscript
- Abstract should include one or two sentence to introduce jasminin.
- Line 57-59: Are the authors referring to exogenously administered TNF-alpha? Clarify in the sentence.
- Line 59-61: Re-word the sentence as it is unclear what authors wants to convey
- Line 62-63: Re-word: ex. Multitude of natural products are known to regulate endogenous cytokine production. Also use appropriate number of references.
- Line 98-100: re-word
- Line 139-142: re-word
- Line 143-144: re word
- Line 148-149: As the authors has not investigated the minimum jasminin concentration required for significant increase in TNF-alpha production, should re-word the sentence.
- Line 151-153:Should include the referred data in the text as supplementary data
- Figure 1B: symbols used to indicate significance does not match between the figure and figure legend
- Line 155-156: Include the timeline of the measurement- 24 hrs.
- Line 169-170: Authors have earlier claimed the IFN-aplha production was unaffected in RAW264.7 cells following treatment with jasminin. please clarify.
- Figure 2A; not necessary include the coloured boxes on the graph as the information is conveyed in the x axis. Use "cell culture supernatant" instead of cell supernatant
- Line 174: Use “cell culture supernatants” instead of “supernatants”
- Line 176-187: re-word
- Line 179-181: Unclear if there is a morphological change in the plaque other than a reduction in plaque numbers. But authors can measure the plaque sizes to present the data more conclusively.
- Line 187-189: re-word
- line 191-195:sentence is not worded correctly
- Lines 203-207: reword the sentences
- Lines 214-216 reword
- Figure 4B: Y axis can be more appropriately labeled. ex. fold increase in phosphorylated protein levels etc.
- Figure 5A: Jasminin+ inhibitors of NFKB- DAPi signal is weak.
Also can calculate the overlap coefficient to better quantify nuclear localization of p65
- Figure 5C: on the x axis of the graph, the BAY11-7082 on the second column is mistakenly marked as +. It should be "0"
- line 257-258: reword
- line 311: authors indicate mechanism of action is different. Not clear on what mechanism action meant in here and the authors have not investigated any mechanism of action. should remove from the sentence.
- line 312-313: re-word
- line 318: refer to the figure 6 in the discussion not in the conclusion.
Reviewer 2 Report
The paper is trying to understand the antiviral effects of jasminin and its related signaling pathways. The following problems must be properly answered and fixed.
1 The authors really need to improve the writing of this paper. There are countless grammar and confusing descriptions in the paper. For instance: There are two types of HSV known 44 as type 1 and 2 which share a similar amino acid similarity. HSV-1 is highly contagious infection that is transmitted through contact with saliva or mucosal surfaces.
Researches showed that TNF-α has a protective effect in acute infection with HSV- 57 1 in mice.
Activated macrophages are known to have a defensive function against viral infection, thus, macrophage activation was assessed by jasminin was investigated.
Not just these mentioned sentences. Please try to improve the language in the whole paper. The current vision is really confusing.
2 As shown in Figure 1C, jasminin stimulated production in a concentration-dependent manner at the concentratations from 3.1 μΜ to 200 μM. Fix the sentence, please.
Can the author explain why a higher concentration of Jasminin shows a low proliferation rate? How does LPS treatment affect the proliferation of RAW 264.7 cells?
Can authors discuss why Jasminin only promotes the production of TNFa but not IFNs?
3 All bar graphs in the paper need to be presented with individual data points. The current version is hard for readers to directly see the replication of experiments.
4 The figure legend in this manuscript needs to be improved. The figure legend should enable the reader to understand a figure without having to refer to the main text of your manuscript. However, there is no detailed explanation of what experiment was done and description for the data in this paper, and the replication of the experiments. For example, in figure2a, the authors should be clear about the cell type used in the experiments.
5 The proper control for figure2c is non-Jasminin treated supernatant. Importantly, the authors claim that “Interestingly, the antiviral effect of jasminin was enhanced in the presence of supernatants containing endogenous TNF-α”. This is not a correct conclusion. How do authors know the antiviral effect comes from Jasminin? Is there any residual Jasminin in the supernatant? What is the half-life of Jasminin?
In figure2C, the positive control 100pg/ml of TNF-a showed the strongest antiviral effect within all the groups. Based on figure1C: “The minimally effective con- 147 centration of jasminin was in a low-concentration of 3.1 μΜ (>200 pg/mL), and maximum 148 TNF-α released in RAW 264.7 cells was about 500 pg/mL at a concentration of 50 μM”. Please explain why 400pg/ml and 500pg/ml of TNFa in the supernatant showed less antiviral effect than 100pg/ml TNFa.
What is the difference between the pure TNFa and the TNFa induced by jasminin? Since the TNFa induced by jasminin showed a weaker antiviral effect than pure TNFa, what is the rationale for using it?
Does positive control TNF-α (100 pg/mL) affect the Vero E6 cell viability?
The sentence “We confirmed that TNF-α has a strong anti-HSV-1 effect, and in addition that the endogenous IFN-α-induced by jasminin was displayed in-direct antiviral effects in vitro.” is confusing. How do authors get to this conclusion? Why IFN-a? What do you mean in-direct?
6 In figure3a, the representative western blotting data of AKT and p-AKT do not come from the same gel blot. Therefore it is hard to confirm if the data is right or not. Please provide representative blotting data obtained from one blot. There is the same problem for figure4 and figure5. Please fix these.
In figure3C, figrue4C and figure5C, 50uM of Jasminin treatment-induced about or more than 1000pg/ml TNFa in the supernatant of RAW264.7. It is different from the result reported in figure1C. Please explain this.
7 What does figure5a try to show? Please explain the data in the results. Why is there less DAPI staining in the Jasminin+inhitiors of NF-kB?
The quantification data for 5B is missing. Please provide it.
8 The paper provided evidence that PI3K/Akt, MAPKs, and NF-κB signaling pathways are involved in the Jasminin mediated antiviral effect, respectively. However, the paper did not provide any direct evidence that the anti-viral effect of jasminin is through classical PI3K/Akt, MAPKs, and NF-κB signaling pathways as shown in figure6. Similar in the discussion. Authors claim “Akt activation by jasminin resulted in the augmentation of NF-κB 287 acetylation in RAW 264.7 cells.” The paper did not show this. To get the conclusion in figure6, the authors need to provide more evidence that perturbing the PI3K/Akt pathway can affect the MAPKs and NF-κB signaling pathways.
9 The discussion is not a real discussion. The authors just repeated the results and tried to overstate their conclusion. Make sure do not overstate your results and your writing is precise enough.
10 The introduction provides some background about HSV infection and TNFa. However, the authors seem to just list all these random facts without putting these facts together in a logical way. The introduction does not provide a good rationale for why the authors did this work. What is the biological significance of this work? What is the gap of knowledge?
Round 2
Reviewer 1 Report
Authors have addressed most of my earlier concerns regarding the manuscript and the manuscript can be accepted at present form
Author Response
Our deepest gratitude goes to you for your careful work and thoughtful suggestions that have helped improve this paper substantially.
Reviewer 2 Report
Thank authors for properly answering my previous questions. After carefully reading the paper, I have the following questions.
1: The authors properly explain why a higher concentration of Jasminin shows a low proliferation rate. I believe this is a point to ponder and readers would be interested to hear from the authors on this point. Please discuss this in the discussion.
2 The authors should discuss the advantage and disadvantages of using endogenous TNFa in the discussion.
3 The description in the results for the figure5A is still missing. Please fix it.
If the experiment was trying to show the nuclear translocation of p65, then the data is not serving this purpose. If there is no DAPI staining, how do authors know there are cells or not in the field? If there are no cells in this field, how do authors know if the P65 signal is not false positive? This figure needs to be replaced with a more convincing figure.
4 In figure3C, figrue4C, and figure5C(now figure5D), 50uM of Jasminin treatment-induced about or more than 1000pg/ml TNF-α in the supernatant of RAW264.7. It is different from the result reported in figure1C. You must be consistent in one paper. This is really important because this is one of the major findings of this paper. The authors replied, “In order to get enough total proteins for western blot assay, large scale and density of cells were seeded. So, the level of TNF-α was different with that in Figure 1C”. This is an interesting answer. In methods, the authors describe the procedure: RAW 264.7 cells were cultured with 25 μM jasminin or inhibitors for 24h. ALL western blotting data in this paper are using the sample of a maximal of 60 mins treated cells. There is no western blotting data using 24h treated cells. The author’s response is confusing and is not convincing.
Please explain and discuss this result in the discussion. The readers must have similar concerns about these results.
5 Although authors claim that activation of NF-κB and MAPK pathways by the PI3K/AKT were well studied. The paper does not provide any direct evidence that activation of NF-κB and MAPK pathways by the Jasminin is through PI3K/AKT. Figure6 is just the author's hypothesis. As a potential mechanism, it can be discussed in the discussion. Or authors need to provide experimental evidence. Figure 6 is overstating the conclusion of the paper. It is misleading and must be revised.
